# Unraveling the Intricate Link: Deciphering the Role of the Golgi Apparatus in Breast Cancer Progression

**DOI:** 10.3390/ijms241814073

**Published:** 2023-09-14

**Authors:** Adrian Vasile Dumitru, Evelina-Elena Stoica, Razvan-Adrian Covache-Busuioc, Bogdan-Gabriel Bratu, Monica-Mihaela Cirstoiu

**Affiliations:** 1Department of Pathology, “Carol Davila” University of Medicine and Pharmacy, 050474 Bucharest, Romania; vasile.dumitru@umfcd.ro; 2Department of Pathology, University Emergency Hospital, 050098 Bucharest, Romania; 3Department of Obstetrics and Gynaecology, University Emergency Hospital, 050098 Bucharest, Romania; monica.cirstoiu@umfcd.ro; 4Department of Neurosurgery, “Carol Davila” University of Medicine and Pharmacy, 050474 Bucharest, Romania; bogdan.bratu@stud.umfcd.ro; 5Department of Obstetrics and Gynaecology, “Carol Davila” University of Medicine and Pharmacy, 050474 Bucharest, Romania

**Keywords:** golgi apparatus, breast cancer, protein glycosylation, protein trafficking, cancer progression, tumor microenvironment, biomarkers therapeutic targets

## Abstract

Breast cancer represents a paramount global health challenge, warranting intensified exploration of the molecular underpinnings influencing its progression to facilitate the development of precise diagnostic instruments and customized therapeutic regimens. Historically, the Golgi apparatus has been acknowledged for its primary role in protein sorting and trafficking within cellular contexts. However, recent findings suggest a potential link between modifications in Golgi apparatus function and organization and the pathogenesis of breast cancer. This review delivers an exhaustive analysis of this correlation. Specifically, we examine the consequences of disrupted protein glycosylation, compromised protein transport, and inappropriate oncoprotein processing on breast cancer cell dynamics. Furthermore, we delve into the impacts of Golgi-mediated secretory routes on the release of pro-tumorigenic factors during the course of breast cancer evolution. Elucidating the nuanced interplay between the Golgi apparatus and breast cancer can pave the way for innovative therapeutic interventions and the discovery of biomarkers, potentially enhancing the diagnostic, prognostic, and therapeutic paradigms for afflicted patients. The advancement of such research could substantially expedite the realization of these objectives.

## 1. Introduction

Camillo Golgi is credited with the identification of the Golgi apparatus, a fundamental organelle inherent to eukaryotic cells [1]. Characterized by its intricate and dynamic nature, the Golgi apparatus is pivotal in an array of cellular activities, predominantly protein modification, segregation, conveyance, and packaging before its designated delivery to specific intracellular locales. Structurally, the organelle comprises overlapping membranous sacs, termed cisternae, which possess a unique architectural design optimized for proficient modification and packaging before routing to defined cellular destinations.

Acting as a central hub, the Golgi apparatus is instrumental in the processing and classification of diverse soluble proteins and lipids, directing them to their intended cellular destinations [2]. Given its seminal position in the secretory continuum, any perturbation in its architecture or functionality can gravely impact cellular protein and lipid equilibrium. Notably, a growing body of research has demonstrated that aberrations in the Golgi apparatus are implicated in a spectrum of conditions ranging from neurodegenerative maladies [3,4,5,6,7] to ischemic strokes, cardiovascular ailments, pulmonary arterial hypertension, infectious diseases, and malignancies.

Intrinsically adaptable, the Golgi apparatus possesses the capacity to rapidly recalibrate in response to evolving cellular demands and extracellular cues. This involves undergoing morphological transitions, such as reorganization, fragmentation, and integration with ancillary organelles, to aptly address variances in protein production and cellular requisites. In this context, the Golgi apparatus engages in complex interplays with other cellular structures, notably the endoplasmic reticulum (ER) and endosomes, orchestrating a sophisticated intracellular transport and communication matrix [8].

The indispensability of the Golgi apparatus in cellular operations underscores the ramifications of its dysfunction on human physiology and health. Disruptions or anomalies in its form, functionality, protein shuttling, or associated metabolic pathways have been pinpointed as etiological agents in a diverse array of pathologies, including malignancies, neurodegenerative diseases, and metabolic anomalies. Consequently, deepening our comprehension of its operational dynamics and significance is imperative for unveiling the underpinnings of these disorders and crafting targeted therapeutic modalities for their efficacious management.

## 2. Golgi Sorting, Protein Trafficking, and Glycosylation Abnormalities

Preserving the structural coherence of the Golgi apparatus is paramount for its optimal operation, as structural perturbations could usher in an array of pathologies. Operational aberrations of the Golgi apparatus encompass modifications in its pH equilibrium, anomalous glycosylation trajectories, and compromised membrane transport. Notably, fragmentation of the Golgi has been postulated as a precursor event in cellular apoptosis [9,10]. In scenarios of pharmacological or oxidative duress, the Golgi apparatus undergoes transformations, such as cargo saturation, ion concentration disequilibrium, and irregular luminal acidity, which collectively can induce membrane transport defects. We have coined the term “Golgi stress” to encapsulate this specific Golgi apparatus response, and two well-discussed molecular pathways are the structural preservation of Golgi apparatus by the TFE3 (transcription factor binding to IGHM enhancer 3) pathway and the proteoglycan pathway, which uptake the expression of enzymes for glycosylation [11].

Glycosylation stands as a pervasive posttranslational modification of proteins and plays a pivotal role in protein-mediated signaling. The glycans situated at glycosylation loci can span a spectrum in terms of complexity, from singular sugar chains to polymers boasting over 200 sugar units. Furthermore, glycans can be subjected to auxiliary modifications, encompassing the addition of entities like phosphate, sulfate, acetate, or phosphorylcholine for further diversification. It is noteworthy that a multitude of glycans manifest branch-like structures. An N-glycan entity can house up to six branches, each embedded with several recurrent disaccharide segments. The work by Stanley et al. (2011) offers insights into the traits and operations of Golgi glycosyltransferases (GTs), encompassing their activity spectra from their initiation at the cis-Golgi to their passage through the trans-Golgi network (TGN) [12].

The glycosylation of proteins is executed at two discrete intracellular locales, each defined by unique attributes. Proteins resident in the cytosol and nucleus undergo O-GlcNAcylation, wherein singular sugar entities termed N-acetylglucosamine (GlcNAc) directly bind to serine or threonine amino acids. This mechanism is instrumental in fine-tuning protein interactions, stability, functionality, and a gamut of cellular undertakings such as transcription, metabolism, apoptosis, and organelle genesis and transport [13,14]. In contrast, within the ER and Golgi apparatus lumen, secretory and transmembrane proteins are subjected to glycosylation by affixing specific glycosaccharides, or glycans, to particular amino acid chains. This modus operandi facilitates their functional diversification, allowing them to partake in multifarious cellular events [15].

The Golgi apparatus houses an array of glycosylation enzymes capable of either cleaving monosaccharides (glycosidases) or attaching them (GTs). Intriguingly, these enzymes can form both heteromeric and homomeric assemblies [16]. Structurally, GTs are membrane-bound proteins characterized by a brief N-terminal segment, a singular membrane domain, and a luminal domain. Due to this intricate configuration, they frequently establish enzyme complexes with other active enzymes within specific glycosylation pathways. N-glycosyltransferases within the Golgi can manifest in either homomeric or heteromeric groupings. The cyclical process of these GTs entails transitions influenced by the microenvironment, oscillating between heteromeric and homomeric states. While homomeric enzyme formations are pivotal in facilitating the folding and transportation of GTs to the Golgi apparatus, the more active heteromers are predominantly utilized for streamlined glycosylation [17,18]. Noteworthy GTs include GalNAc-T2 (N-acetylgalactosaminyltransferase-2) and GalT (β1,4-galactosyltransferase), which, due to their specificity for the Golgi apparatus, highlight that any depletion of juxtanuclear Golgi staining might be indicative of the organelle’s attributes and the associated membrane proteins [19].

A dysfunctional Golgi glycosylation process has been associated with invasive behavior in various cancer types, encompassing prostate and breast malignancies [20,21]. The glycosylation process within the Golgi plays a cardinal role in numerous oncogenic molecular and cellular sequences, such as signal transduction, cellular communication, dissociation and invasion of cancer cells, cell–matrix attachment, angiogenesis, immunomodulation, and metastasis [22]. Analogous to the function of epithelial cadherin in mediating epithelial cellular cohesion, the Golgi-mediated glycosylation of N-linked glycans on epithelial cadherin might influence the epithelial-to-mesenchymal transition, thereby catalyzing the emergence of metastatic outgrowths. Such a mechanism is postulated to facilitate the migratory capacity of neoplastic cells from their inception point, be it during reparative processes post-injury or other standard physiological events, and becomes instrumental in the metastatic spread and proliferation of cancer [8,23].

The GOLPH3 complex, recognized as Golgi phosphoprotein 3, stands as a pivotal molecular entity in the realm of Golgi-facilitated oncogenesis. Its centrality in cancer can be attributed to a myriad of critical functionalities. GOLPH3 not only orchestrates Golgi glycosylation pivotal for the cancerous phenotype manifestation but also amplifies the DNA (deoxyribonucleic acid) damage response, bolstering survival amidst DNA-injurious scenarios. Additionally, it synergizes with retromer elements to enhance the mTOR (mammalian target of rapamycin) signaling upon growth factor induction and facilitates cell motility by orienting the Golgi apparatus toward the cellular forefront. Beyond GOLPH3, the Golgi spectrum hosts another consequential protein, GM130 (Golgi matrix protein 130). Integral to Golgi glycosylation and membranous protein trafficking, the downregulation of GM130 culminates in autophagy, diminished angiogenesis, and suppressed tumorigenesis [6,23,24,25].

Dysregulated Golgi glycosylation not only holds implications for carcinogenesis but might also propel cancer progression. Given the intertwined nature of Golgi-related operations and oncology, delving into and therapeutically targeting these processes should be foundational in cancer research endeavors.

Divergences in glycosylation can engender alterations in the conformation and function of numerous membranous proteins, with particular significance to collagen, fibronectin, integrins, and laminin at the extracellular interface. The paramount role of transmembrane integrins lies in fortifying the cytoskeleton via myriad cell–cell and cell–matrix interactions, thereby catalyzing cellular maturation and proliferation [26]. Glycosylation aberrancies might culminate in the flawed anchorage of these proteins, engendering a plethora of pathologies encompassing neurodegenerative conditions, malignancies, and cardiovascular afflictions [27,28,29].

The Golgi apparatus, with its cardinal role in modulating core cellular mechanisms, like adhesion and migration, stands as a keystone in the panorama of cancer evolution and metastatic dissemination. A prominent influencer in these oncogenic processes is identified as phosphatidylinositol 4-phosphate (PI4P). Hence, its role in human breast cancer can markedly sway cell–cell adhesion and migratory patterns [24].

Recent investigations underscore the paramount regulatory role of PI4P in the structural and functional intricacies of the Golgi apparatus, notably affecting glycosylation and the trafficking of proteins pivotal to cell–cell adhesion. By modulating the Golgi PI4P concentrations, the localization and activity of cardinal adhesion molecules, such as E-cadherin, are affected, thereby reshaping the intensity and dynamics of cell–cell interactions. Beyond its role in adhesion, PI4P governs activities linked with enzymes crucial for the synthesis or restructuring of glycosphingolipids, which are indispensable for cell surface interactions and signaling modalities. Furthermore, PI4P is integral in governing invasive cellular motility, a critical phenomenon in oncologic metastasis. Its regulatory role in Golgi-centric vesicular trafficking and membranous dynamism facilitates the modulation of invasive cell polarization and protrusive activities, augmenting their migratory and invasive propensities. In this orchestration, PI4P collaborates with a cohort of Golgi-associated proteins and lipid-mediated signaling pathways to modulate cytoskeletal transformations and matrix degradation, thereby facilitating the metastatic voyage of cancerous cells [21,30].

The traversal of cargoes through the Golgi apparatus is a multifaceted event and remains a focal point of discourse in the scientific literature. This review sheds light on five contemporary models postulated for assessing Golgi traffic, weighing their merits and demerits. The inaugural model posits anterograde vesicular transport amidst stable compartments of the Golgi. Conversely, the second hypothesis advocates for cisternal progression/maturation, wherein Golgi cisternae transition through sequential maturation phases. The third paradigm fuses progression/maturation with heterotypic tubular conveyance between cisternae. The penultimate model champions swift protein partitioning within a heterogenous Golgi and the terminal model envisions stable compartments as precursors for ensuing cisternal development.

A meticulous analysis reveals that no singular model can holistically encapsulate all documented phenomena across varied organisms. It might be more tenable to perceive cisternal progression/maturation as a foundational and evolutionarily conserved mechanism governing Golgi traffic. Certain cellular systems might integrate heterotypic tubular transport within Golgi cisternae. A judicious exploration of these models will illuminate the intricate facets of Golgi traffic, bestowing deeper insights into its operational mechanisms and elucidating this quintessential cellular undertaking. Grasping its foundational tenets is indispensable for decoding its influence on cellular equilibrium as well as pathological states linked to protein trafficking or excretion [31].

## 3. Golgi Apparatus Involvement in Breast Cancer

Breast cancer, a pressing global health challenge, leads to female mortality rates, and its prevalence is anticipated to surge in the forthcoming years. Diagnostic techniques like mammography and clinical breast inspections are pivotal for its early identification. While therapeutic modalities encompass surgical interventions, chemotherapy, and radiation treatments, each come with a set of concerns. Chemotherapy, despite its efficacy in neutralizing cancerous cells, presents a suite of adverse reactions. Radiotherapy, typically paired with surgery, may inflict enduring harm to critical organs. More promising therapeutic avenues encompass the deployment of anti-ErbB2 antibodies, exemplified by trastuzumab, especially for HER2-positive breast cancer variants. Additionally, antiestrogens and aromatase inhibitors serve to suppress the manifestation of estrogen-associated genes, proffering treatment avenues with diminished side effects [32,33].

The Rab GTPases, pivotal orchestrators of vesicular transportation, hold profound implications for the malignancy and invasiveness of cancer cells. Delving into estrogen receptor-positive breast cancer cellular frameworks reveals the instrumental role of Rab27B. Its heightened expression correlates with an augmented cellular elongation and an escalated invasiveness when interacting with collagen matrices. Such effects can be counteracted through miRNA-mediated interventions. Moreover, the amplification of Rab27B expression bears a direct relation to the surge in HSP90 alpha expression, a molecular custodian pivotal for upholding the structural integrity of MMP2 [34,35].

Rab40B’s influence is palpably seen in maneuvering the trafficking pathways of metalloproteases MMP2 and MMP9 within the MDA-MB-231 breast cancer cellular context, facilitating the degradation of the external cellular matrix. Another metalloprotease, MT1-MMP, falls under the regulatory domain of Rab2A, which further fuels metastatic behaviors via its interaction with the VPS39 protein and is crucial for the amalgamation and clustering of late endosomes/lysosomes.

SiRNA screening has unmasked Rab2A’s regulatory influence over the Golgi transport mechanisms of surface E-cadherin in breast cancer cells. Given the pivotal role of E-cadherin loss as an oncogenic transformation indicator, these revelations accentuate the significance of Rab GTPases in dictating vesicular transportation mechanisms. Such processes wield influence over cellular structural dynamics, invasion capacities, and external matrix degradation in these particular breast cancer cells [36,37]. Refer to Figure 1 for a visual representation.

As highlighted in the cited study [38], GOLPH3’s pronounced overexpression in breast cancer cells and tissues contrasts starkly with its presence in normal breast tissue. Escalated GOLPH3 levels correlate with advanced tumor development, metastatic spread, and a grim prognosis for breast cancer sufferers.

In the breast cancer scenario, GOLPH3 emerges as a pivotal entity, underpinning cancer cell proliferation and longevity by modulating its DNA damage response apparatus. A noteworthy interaction of GOLPH3 is with ATM (ataxia-telangiectasia mutated), a quintessential DNA damage response protein located at the Golgi. This interaction amplifies survival rates, rendering cancer cells more resilient against DNA damage-driven cellular demise. Such a mechanism equips cancer cells with heightened resistance against the genotoxic assaults unleashed by treatments like chemotherapy and radiation [39].

Furthermore, GOLPH3’s tentacles extend into cancer progression by exerting regulatory control over a slew of signaling pathways, notably the PI3K-AKT-mTOR axis. GOLPH3 bolsters AKT activation, a kinase pivotal for cell proliferation and survival. This ultimately accelerates tumor growth and endows them with fortified resistance against treatments. From a therapeutic lens, targeting GOLPH3 emerges as a promising stratagem in the battle against breast cancer. A nuanced inhibition of its expression or functional prowess could prime cancer cells for increased susceptibility to DNA-damaging agents, effectively crippling tumor proliferation and metastatic spread [40].

In breast cancer patients, a surge in gene expression linked to ER-Golgi transport processes is evident, exemplified by genes like ARF4, COPB1, and USO1. These genes play an instrumental role in ferrying proteins between the ER and Golgi apparatus. To elucidate further, COPII vesicles shepherd proteins from the ER to the Golgi, whereas ARFs pilot the retrograde journey from the Golgi to the ER, which is facilitated by COPI vesicle formation [41].

Delving into the transportation dynamics of these genes reveals intriguing insights. The overexpression of ARF4, COPB1, and USO1 accelerates protein shuttling from the ER to Golgi. Introducing biotin amplifies this trafficking tempo even more, hinting at the pivotal role these ER-Golgi trafficking genes play in optimizing transportation kinetics [41].

A comprehensive meta-analysis of breast cancer cells also unravels that the expression patterns of ARF4, COPB1, and USO1 are orchestrated by the CREB3-like transcription factors. This harmonious co-expression, when disrupted, wreaks havoc on the cellular adhesive capacities, mobility, invasion potential, and the overarching metastatic traits of cancerous cells [32].

The discoveries highlight the pivotal roles that ARF4, COPB1, and USO1 undertake in breast cancer cell proliferation and invasiveness. Their significance underscores their role as key contributors to the disease’s progression. Their paramount importance is further spotlighted through their integral roles in breast cancer evolution, particularly via the ER-Golgi trafficking mechanisms [42].

CREB3, a transcriptional architect, is instrumental in governing the traffic between the ER and Golgi apparatus. Its implications for breast cancer metastasis are a subject of fervent research. The quest to understand gene expression footprints steered by CREB3-mediated ER-Golgi trafficking unveils repercussions on the metastatic journey of breast cancer [43].

Evidence affirms that CREB3 activation spurs the upregulation of genes that participate in ER-Golgi trafficking, prominently featuring constituents of the COPII and COPI vesicle transportation networks. This unique genetic footprint, orchestrated by CREB3, is tethered to enhance metastatic capabilities in breast cancer cells. A slew of experimental methodologies was deployed in this research, which strove to pinpoint the direct nexus between CREB3-driven trafficking and the invasiveness inherent to breast cancer cells [44].

Moreover, the CREB3-directed trafficking signature has been painted as a harbinger of grim clinical outcomes in breast cancer patients. An upsurge in the expression of signature genes coincides with an elevated risk of metastasis and a dip in overall survival rates. This spotlight on the clinical significance accentuates its potential as a harbinger of disease prognosis in breast cancer [21].

In addition, genetic and epigenetic deviations in loci affiliated with the Golgi apparatus, such as CCDC170 (Coiled–Coil Domain Containing 170), have been entwined with susceptibilities to breast cancer. The unraveling of Golgi microtubule organization by proteins, like CCDC170, can culminate in anomalies in cell polarity and motility. These facets are quintessential for the invasiveness and metastatic prowess inherent to cancer cells [45].

## 4. Estrogen-Mediated Regulation of Protein Transcriptome: Impact on Vesicular Trafficking and Giant Vesicle Formation in Breast Cancer

Giant Vesicles (GVs) are vesicles, either inside or outside cells that range in size from 3 to 42 µm amd play a pivotal role in tumor proliferation. These vesicles originate mainly from ERα (estrogen receptor alpha)-negative breast cell lines and predominantly reside at the cell’s edge [46].

Estrogen, pivotal in steering the transcriptome of proteins involved in vesicle movement and GV formation in breast cancer cells, regulates gene expressions essential to these processes. By modifying this transcriptome, estrogen directs the creation of vital proteins for vesicle movement and GV formation, thereby fueling growth and disease progression. As such, estrogen stands as a chief architect in this molecular realm and is linked with vesicle movement and GV formation. Estrogens, as paramount female sex hormones, are intrinsic to many physiological and pathological functions and hold a significant role in the onset of breast cancer. They operate by docking onto nuclear estrogen receptors, subsequently reshaping gene expressions. Notably, genes dictated by estrogen have been identified to sway various dimensions of cancer cell movement [47,48,49].

Two gene standouts, SYTL5 (Synaptotagmin-like 5) and RAB27B, regulated by 17 ss estradiol, are central to vesicle movement and exocytosis. SYTL5 functions as an intermediary molecule, collaborating with GTPases RAB27A/B. An increased presence of these genes has been documented in estrogen receptor-positive breast cancer cell lines, emphasizing their association with vesicle movement [50].

Additionally, SNX24 (Sorting Nexin 24), another estradiol-influenced gene, plays a quintessential role in endosomal categorization. GALNT4 (Polypeptide N-Acetylgalactosaminyltransferase 4) and SLC12A2 (Solute Carrier Family 12 Member 2) are foundational to the vesicle movement mechanism. Specifically, SLC12A2 is instrumental in steering the exocytosis of catecholamine from chromaffin cells, governing breast shape dynamics [51].

Wright et al. unveiled a unique vesicle species in breast cancer cells, which is known as the GV. Breast cancer cells, like MCF 7 and T47D, which robustly express the estrogen receptor alpha, rely on estradiol for their genesis. In contrast, non-ER alpha-negative cells, like MDA-MB-231/MDA-MB-468, remained untouched by estradiol in GV formation. However, in the presence of ER alpha, estradiol instigated the inception of estradiol-dependent GVs, hinting at a possible route where estradiol might spark this formation through ER alpha expression [52].

In essence, genes guided by estradiol and their interplay in vesicle movement across diverse frameworks considerably drive breast cancer cell growth and spread. This provides a compelling narrative on the multifaceted relationship between estrogen cues, gene orchestration, and vesicle movement in breast cancer. These discoveries elucidate the sophisticated dance between estrogen signaling, gene oversight, and vesicle movement, shedding light on their role in the progression of the disease [53].

## 5. Inhibition of Golgi-Associated Lipid Transfer Proteins (LTPs) as Potential Targets for Disease Intervention

The Golgi complex (GC) is pivotal in lipid biosynthesis and distribution. This incorporates both vesicle transport and non-vesicular pathways via Lipid Transfer Proteins (LTPs) like CERT (ceramide transfer protein), OSBP (oxysterol-binding protein), and FAPP2 (four-phosphate adaptor protein 2). Each of these proteins boasts distinct transport capabilities: CERT shuttles ceramide, OSBP transfers cholesterol, and FAPP2 moves GlcCer. All these proteins carry an N-terminal PH domain, enabling them to bind with PI4P for effective delivery within the GC. Some inhibitors targeting these processes are emerging as potential antiviral and anticancer therapeutics [54].

CERT specializes in carrying ceramide from the ER to the TGN, where it undergoes conversion to sphingomyelin [55]. Its inhibition or depletion results in ceramide accumulation, catalyzing ceramide-induced ER stress. This phenomenon primes various cancer cells, including ovarian, colorectal, and HER2-positive breast cancer cells, for enhanced vulnerability to chemotherapy. HPA-12, a CERT inhibitor, works by hampering CERT’s recruitment during viral or parasitic invasions. This leads to augmented ceramide levels, rendering cancer cells more susceptible to paclitaxel-induced cell death, especially in contexts where these cells are resistant to traditional paclitaxel treatments or when there are infections or parasitic interferences [56].

OSBP1, a specialist in binding oxysterol, is central to the swap of cholesterol for PI4P between the ER and GC. OSBP, alongside ORP4L (oxysterol binding protein (OSBP)-related protein 4L), has been pinpointed as a target for various anticancer agents, including ORPphilin compounds like cephalostatin 1, OSW-1, Ritterazine B, and Schweinfurthin A, due to their profound effects on lipid metabolism. Moreover, Itraconazole, primarily an antifungal, demonstrates anticancer efficacy by targeting OSBP. Summarily, the GC’s equilibrium in lipid concentrations hinges on both vesicle-based transport and the action of lipid transfer agents, like CERT and OSBP. Strategies that target these proteins are emerging as promising avenues in the development of novel anticancer and antiviral treatments [57].

## 6. Rho-Related BTB Domain Containing 1 (RhoBTB1) Drives Breast Cancer Growth and Metastasis through Methyltransferase-like 7B (METTL7B) Regulation

In breast cancer cells and patient samples, there is a marked downregulation of RhoBTB1 (Rho-Related BTB Domain Containing 1) expression, hinting at its probable tumor suppressor properties. Scientific inquiry reveals that a lack of RhoBTB1 leads to the disintegration of the Golgi structure, compromising its functions. This results in anomalies in protein glycosylation and associated trafficking pathways [58].

Adding another layer, METTL7B (Methyltransferase-like 7B) operates downstream of RhoBTB1. Recognized for its role in protein glycosylation, METTL7B experiences an uptick in its expression in breast cancer cells. Its heightened presence is, unfortunately, an indicator of grim outcomes for cancer patients. Through various functional studies, it is discerned that a surge in METTL7B expression counteracts the invasive tendencies of breast cancer cells and concurrently rectifies the Golgi disintegration brought on by RhoBTB1 insufficiency [59].

This interplay implies that RhoBTB1 acts as a brake on METTL7B, ensuring the structural and functional integrity of the Golgi apparatus and curtailing the invasion of breast cancer cells. Disturbances in this delicate balance seem to facilitate aggressive tendencies in breast cancer cells [60].

The narratives surrounding RhoBTB1 and RhoBTB2, and their associations with breast cancer, are subjects of intrigue, but their precise roles remain shrouded in mystery. RhoBTB3’s part in breast cancer deterrence is even less understood. To delve deeper into these ambiguities, this research utilized bioinformatics tools, like Oncomine and cBioportal, and aimed to elucidate the roles and potential prognostic value of RhoBTB3 and Col1a1 in the context of breast cancer. Comprehensive methodologies including qRT-PCR analysis, immunoblotting assays, and a range of functional tests, such as invasion and proliferation assays, and flow cytometry were employed. The endgame was to pinpoint their contribution to the trajectory of breast cancer and evaluate their worth as prognostic indicators [61].

## 7. Possible Therapeutic Targets Regarding Breast Cancer

Breast cancer presents itself as a multifaceted ailment that is characterized by diverse subtypes and molecular variations. This heterogeneity necessitates a wide spectrum of treatment approaches targeting different pathways and molecules (Table 1). A breakdown of these targeted pathways and molecules follows.

In estrogen receptor-positive breast cancers, the cancer cells thrive on estrogen signals. To curtail this growth, endocrine therapies are deployed. Notable examples include selective estrogen receptor modulators, like Tamoxifen, and aromatase inhibitors, such as Letrozole, which have proven adept at inhibiting cancer cell proliferation [62].

Breast cancers characterized by excessive HER2 expression are labeled “HER2-positive”. This overexpression catalyzes rapid cell growth and survival. Targeted therapies, specifically designed to obstruct HER2 signaling, are the countermeasures employed here. Drugs like trastuzumab (Herceptin), pertuzumab (Perjeta), and ado-trastuzumab emtansine (Kadcyla) are paramount in battling these types of breast cancers [63].

When it comes to both estrogen receptor-positive and HER2 breast cancers, there is an interesting dynamic at play involving SCAMP1 (Secretory Carrier-Associated Membrane Protein 1). Acting in post-Golgi recycling pathways, this protein, deemed a tumor suppressor, enhances the trafficking of MTSS1 (metastasis suppressor protein 1) to the cell’s surface. Once there, MTSS1 kickstarts Rac1-GTP, promoting cell–cell adhesion, thereby forming a defensive front against cancer progression and invasion [64].

Lastly, for breast cancers carrying mutations in the BRCA1/2 genes, Poly (ADP-ribose) Polymerase (PARP) inhibitors come into play. Drugs like olaparib, talazoparib, and niraparib capitalize on DNA repair deficiencies in these cancers. By targeting cells that show homologous recombination inadequacy, they demonstrated significant clinical benefits for patients with BRCA mutation-positive breast cancer cases [65].

Breast cancer’s complexity necessitates a wide array of targeted therapeutic interventions to tackle its multifarious molecular pathways. Several key molecules and pathways in breast cancer therapy include:

Cyclin-Dependent Kinase 4/6 (CDK4/6)—Drugs such as palbociclib, ribociclib, and abemaciclib, have been developed to inhibit CDK4 and CDK6 activities, which are pivotal for the progression of the cell cycle. Their introduction has yielded promising results, especially for patients with hormone receptor-positive and HER2-negative advanced breast cancer, showcasing significant improvements in progression-free survival [66].

The PI3K/AKT/mTOR Pathway—A recurrent aberration observed in breast cancer patients is a malfunction in this particular signaling pathway. As a consequence, inhibitors targeting its constituents, like PI3K inhibitors (e.g., alpelisib) and mTOR inhibitors (e.g., everolimus), have emerged as particularly potent against certain breast cancer variants [67].

Immune Checkpoint Inhibitors—These are groundbreaking agents, such as pembrolizumab and atezolizumab. They are engineered to target immune checkpoints, notably PD-1/PD-L1 (Programmed Cell Death Ligand 1), amplifying the immune system’s capacity to identify and destroy cancer cells. Their efficacy has been particularly notable in cases of advanced triple-negative breast cancer [68].

Moreover, it is worth highlighting that the Golgi apparatus has sparked interest as a potential molecular target in breast cancer therapy. A growing number of therapeutic avenues, including antibody–drug conjugates, nanoparticles equipped to deliver antibody drugs, conjugate drug therapies, and advanced immunotherapies, are under scrutiny to optimize cancer patient outcomes [69]. The selection of a treatment strategy is contingent on myriad factors, including the patient’s tumor subtype, its stage, and substage classification.

## 8. Conclusions

Breast cancer’s progression and initiation intricately intertwine with the functioning of the Golgi apparatus. Alterations in its structural and operational framework have deep implications, manifesting in the acceleration of tumor growth, invasive behavior, and metastasis. A significant causative factor is the malfunctioning of Golgi-associated proteins, like matrix proteins, lipid transporters, and resident enzymes, leading to anomalies in glycosylation, protein trafficking, and signal relay—fundamental hallmarks of breast cancer.

CCDC170 stands out as a critical genetic component interlinked with the Golgi system with profound effects on breast cancer proliferation. Any perturbations within this structure might trigger cellular transformations, escalating the risk of cell invasion and potential metastasis.

Golgi-associated proteins, extending from matrix proteins to lipid transfer agents and enzymes resident within the Golgi, are complicit in fostering tumor growth, invasiveness, and metastatic spread. They modulate vital cellular functions, including adhesion, migration, and proliferation—fundamental processes underpinning the progression of breast cancer.

The intricate nexus between the Golgi apparatus and breast cancer is a treasure trove of insights. It reveals the underlying molecular intricacies and illuminates potential therapeutic interventions. By zeroing in on specific Golgi-mediated processes, such as protein glycosylation, vesicle transport, and interactions between Golgi and microtubules, we unveil novel therapeutic horizons. These might halt tumor expansion, avert metastatic spread, and enhance the prognosis for breast cancer patients.

However, the enigma of the Golgi apparatus and its relationship with breast cancer demands further exploration. By deciphering its elaborate processes and their role in cancer evolution, we are better poised to develop advanced diagnostic instruments, tailor treatments more precisely, and ensure a brighter prognosis for those battling breast cancer.

## Figures and Tables

**Figure 1 ijms-24-14073-f001:**
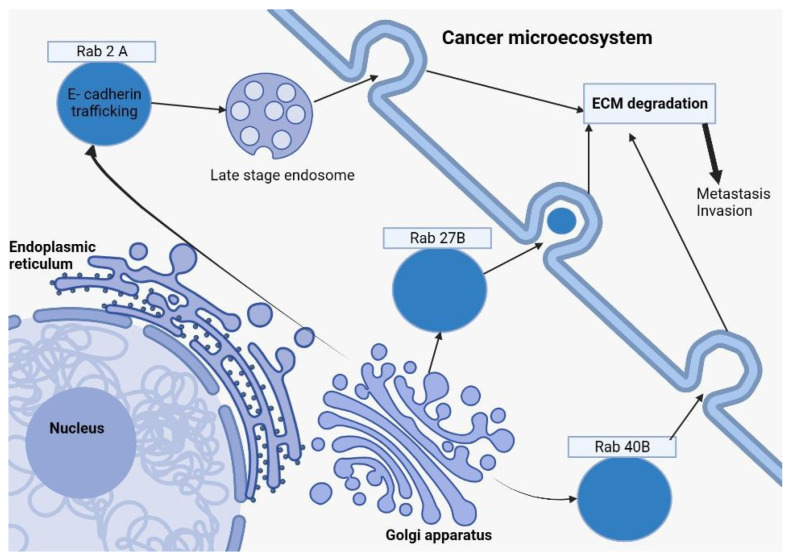
This diagram depicts how Rab27B, Rab2A (in a late-stage endosome form), and Rab40B from the Rab family of proteins, formed by the Golgi apparatus, are involved in breast cancer. After exocytosis, those Rab proteins play a key role in ECM (Extracellular Matrix Degradation), a representative element in the cancer microecosystem, which will further determine cell proliferation, tumoral invasion, and the formation of metastatic masses.

**Table 1 ijms-24-14073-t001:** Pharmaceutical agents for breast cancer treatment with Golgi apparatus implications.

Type of Breast Cancer	Drug Name	Development Stage	Golgi Apparatus Mechanism
Estrogen receptor-postive	Tamoxifen	FDA-approved	During post-Golgi recycling pathways, SCAMP1 determines cellular modifications against the progression of cancerous processes
	Letrozole	FDA-approved	
HER2-positive	Trastuzumab	FDA-approved	
	Pertuzumab	FDA-approved	
	Adotrastuzumab emtansine	FDA-approved	
HER2-negative	Palbociclib	FDA-approved	Trafficking HER2 from the Golgi to the endocytic cellular pathways leads to a decreased level of HER2 expression
	Robociclib	FDA-approved	
	Abemaciclib	FDA-approved	
BRCA1/2	Olaparib	FDA-approved	Inhibition of Poly (ADP-ribose) Polymerase (PARP)
	Talazoparib	FDA-approved	
	Nitraparib	FDA-approved	
Triple-negative	Pembrolizumab	FDA-approved	AXL (tyrosine kinase) has a polarized localization at the Golgi apparatus
	Atezolizumab	FDA-approved	

## Data Availability

All data are available online in libraries such as PubMed.

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
