# Peer review of "Unraveling the Intricate Link: Deciphering the Role of the Golgi Apparatus in Breast Cancer Progression"

_ijms, 2023, doi:10.3390/ijms241814073_

Round 1

Reviewer 1 Report

In this manuscript, the authors summarized functions of Golgi apparatus.  The review is well-organized and covers the topic in protein modification and trafficking, cellular activity regulation, health conditions especially breast cancer progress, etc. There is one concern to be addressed prior to publication as below:

In line 348-349 on page 8, the title doesn’t match content of this section.

Author Response

Dear Reviewer,

Thank you for your positive feedback and kind suggestions,

We have changed the title of lines 348-349 on page 8

Thank you for your significant contribution!

Reviewer 2 Report

Dear Authors,

I am writing to express my appreciation for the manuscript titled "Deciphering the Role of the Golgi Apparatus in Breast Cancer Progression. I would like to offer a few minor suggestions which further enhance your work.

1. In lines 67-68, you have introduced the concept of Golgi stress. However, it would be beneficial if you could elaborate on the specific GA response associated with Golgi stress.

2. On line 69, you mentioned "posttranslational alterations." I suggest using the more widely recognized term "posttranslational modifications" for greater clarity and alignment with existing terminology.

3. In line 404, you reference the drugs Palbociclib, Ribociclib, and Amebacicilib. It would be clearer if you substituted the term "Instruments" with "drugs" to accurately represent their nature.

4. For the credibility of your claims and findings, I recommend incorporating additional citations and references into your article.

I appreciate your valuable contribution to the breast cancer research field.

There are a few typos and English grammar errors that should be rectified.

Author Response

Dear Reviewer,

Thank you for your positive feedback and guidance!

I’ve added a brief summary of the main molecular pathways associated with Golgi stress

I’ve changed the term “posttranslational alteration” to “posttranslational modifications”

I’ve now substituted the term “instruments” with “drugs”

Moreover, a comprehensive table with current drugs tested for breast cancer was added

Thank you for your significant contribution!

Reviewer 3 Report

Review "Unraveling the Intricate Link: Deciphering the Role of the Golgi Apparatus in Breast Cancer Progression" by Dumitru et al.

The manuscript effectively delves into the multifaceted functions of the Golgi apparatus, highlighting its pivotal role beyond its fundamental cellular functions. The discussion astutely explores its involvement in breast cancer, shedding light on a less explored aspect.

One suggestion to enhance the manuscript would be for the authors to provide a comprehensive list of pharmaceutical agents currently either undergoing clinical trials or approved by the FDA for breast cancer treatment. Furthermore, it would greatly benefit the readers if the authors could expound upon how these therapeutic agents exploit the Golgi apparatus in their mechanisms of action. This addition would enrich the understanding of the intricate interplay between the Golgi apparatus and the therapeutic targets in the context of breast cancer.

Author Response

Dear Reviewer,

Thank you for your positive feedback and guidance!

I’ve added a comprehensive table with current drugs tested for breast cancer, with their possible Golgi apparatus implications 

Thank you for your significant contribution!